# The Effect of Fluid Restriction and Intake Conditions on the Shooting Performance of Competitive Adolescent Handball Players

**DOI:** 10.3390/nu16234246

**Published:** 2024-12-09

**Authors:** Erdem Uylas, Egemen Mancı, Nidia Rodriguez-Sanchez, Cem Şeref Bediz, Erkan Günay

**Affiliations:** 1Department of Physical Education and Sports, Dokuz Eylul University, 35330 Izmir, Turkey; erdemuylas1995@gmail.com; 2Department of Exercise and Sport Sciences, Izmir Demokrasi University, 35140 Izmir, Turkey; egemenmanci@gmail.com; 3Physiology, Exercise and Nutrition Research Group, University of Stirling, Stirling FK9 4LA, UK; nidia.rodriguezsanchez@stir.ac.uk; 4Faculty of Medicine, University of Kyrenia, 33000 Mersin, Turkey; cem.bediz@gmail.com; 5Department of Coaching Education, Manisa Celal Bayar University, 45040 Manisa, Turkey

**Keywords:** fluid restriction, fluid intake, shooting performance, adolescent, handball

## Abstract

(1) Background: This study aimed to investigate the effects of fluid restriction and intake (water vs. sports drink) on shooting accuracy and speed in adolescent handball players, a population with high sensitivity to hydration levels yet understudied in this context. (2) Methods: A total of 47 adolescent competitive handball players (15.04 ± 1.5 years) were included in this study, and the participants were divided into low, average, and high performance according to their shooting performance in the familiarization session. All participants were exposed to fluid restriction and intake conditions during handball training on different days. Before and after the training sessions, changes in shooting accuracy and speed were evaluated. (3) Results: The training protocol resulted in body mass loss in the Average Performer group (*p* = 0.001). Compared to fluid restriction, fluid intake (water intake and sports drink intake) post-training had a positive main effect on shooting accuracy, F(2, 88) = 34.32, *p* < 0.001, η_p_^2^ = 0.44, and shooting speed, F(1, 61) = 4.05, *p* = 0.35, η_p_^2^ = 0.84. (4) Conclusions: Hydration level plays an important role in shooting accuracy and speed performance in adolescent handball players. Therefore, fluid intake integrated into training or match sessions may contribute to the maintenance and improvement of shooting performance.

## 1. Introduction

Inadequate fluid intake or fluid restriction processes lead to core temperature values rising to ~40 °C in athletes [1]. Increased core temperature causes physiological stress and a gradual decrease in body water through sweating rate. Hypohydration, defined by a 1–2% decrease in body weight during exercise, is widely acknowledged as the threshold at which performance impairments in coordination, cognitive function, and reaction time occur [1,2]. Previous study reports indicate that adolescent athletes can only meet 2/3 of their Body Mass (BM) fluid needs if they do not prevent body water loss during training and competition [3]. This situation results in the need to regulate fluid intake processes in adolescent athletes and to examine the effects of different fluid intake types and processes on different performance outcomes, especially under dehydration conditions. Few studies have examined fluid intake processes under exercise-associated dehydration conditions in the adolescent population.

High-intensity and intermittent sports activities and sweat losses caused by an increased core temperature reduce body water by 1.0–1.6 L per hour (h). These processes are most common in shooting-based sports such as handball, basketball, tennis, and cricket [4,5]. In the literature, studies evaluating the shooting performance of adolescent athletes under hypohydration conditions showed that adolescent tennis players had decreases in service accuracy (~30%) and stroke accuracy (~69%) [6]. In another study [7], it was reported that the shooting accuracy and shooting speed of adolescent cricket players decreased in hypohydration conditions, and similar hypohydration conditions negatively affected shooting accuracy in basketball players [8]. In addition, studies recommend fluid intake such as water intake, sports drink intake, etc. in terms of general health and performance decline.

The purpose of fluid intake is to prevent the performance-impairing effects of hypohydration by replacing the fluid lost during exercise with preferred beverages in specified periods and to increase performance [1,2,3]. In this process, water intake creates a stabilizing effect, but it is less preferred by athletes due to its tastelessness. Flavored sports drinks prepared with carbohydrates and electrolytes are preferable [1]. When the relationship between shooting accuracy and fluid intake is considered, water intake maintains shooting accuracy and even provides low-level (3–4%) increases [9]. Hypohydration (−1 to 2% BM) induced by fluid restriction significantly negatively affects cognitive-motor task performance. This impairs motor control and thus worsens shooting performance [10]. Sports drinks contribute to performance increases by delaying physiological deterioration with carbohydrates and electrolytes in their contents [11,12]. Adolescents have smaller body surface area-to-mass ratios and variable sweat rates compared to adults, which requires specialized hydration approaches, especially in high-intensity sports such as handball [1]. Carvalho et al. in their study on fluid restriction in adolescent basketball players showed that shooting accuracy and speed did not change under hypohydration conditions, but performance improved with the consumption of sports drinks (carbohydrate solution) [3]. In their review, Barnes et al. reported that carbohydrate-containing sports drinks consumed during training sessions indirectly increased shooting accuracy at the end of training [13].

Although the number of adolescent handball players in Europe is approximately one million [14,15], the relationship between fluid intake, fluid loss, and performance in this population has been discussed to a limited extent. In particular, there are no studies on the effect of fluid intake on shooting performance in adolescent handball players. Shooting accuracy and speed affect handball game results. Therefore, it is important to investigate the parameters (accuracy and speed) affecting shooting performance. For these reasons, this study aimed to investigate the effects of fluid restriction and different types of fluid intake on shooting performance and shooting speed in adolescent handball players.

The main hypotheses of this study are as follows: (i) fluid restriction during training may impair shooting accuracy and speed in adolescent handball players, (ii) water and sports drink intake may positively affect shooting accuracy and speed, and (iii) fluid restriction and fluid intake protocols may affect shooting accuracy and speed at different levels according to shooting performance level.

## 2. Materials and Methods

### 2.1. Study Design

The study was conducted over four sessions (Figure 1) including one familiarization session and three experimental intervention (fluid restriction, water, and sports drink intake) conditions. A single research group was included in each of these three conditions. Participants who were divided into low–average–high performance groups according to the shooting performance groups in the familiarization condition were included in the fluid restriction, water, and sports drink groups. The study was planned with a randomized crossover design so that the participants were evaluated in all conditions. In all sessions, temperature and humidity were controlled at 28.8 ± 0.5 °C and 45.5% humidity, typical of competitive environments, to ensure ecological validity in simulating match conditions. BM measurements were taken at the beginning of the fluid restriction, water, and sports drink intake sessions. In the warm-up period (total ten minutes), low-to-moderate running and basic passing technique drills followed by dynamic stretching exercises with upper extremity intensity were performed. Then, the shooting performance test was started. Afterward, the participants had to shoot to make 10 shots on the plate hung on the handball goal, while remaining stationary on a 7 m handball line (Figure 1). Accuracy rate and speed were recorded simultaneously during the shots using a Bushnell radar gun and a recognized gold-standard radar gun (JUNGS, Model: 101911, CA, USA). After shooting performance test, a 60 min handball training program was applied to the participant group. In the water intake and sports drink intake sessions, participants were asked to consume 1% of their BM at five-minute intervals. This rate was calculated by using the body mass loss rate per 60 min of handball training during the familiarization session and the study of Rodrigez et al. [16]. During the water and sports drink intake sessions, participants were asked to consume 1% of their BM in five minutes. Especially for sports drink intake, 5 min intervals were preferred in both conditions to facilitate consumption and reduce potential risks such as sudden increases in blood glucose levels [17,18]. The fluid levels that the participants should consume were determined in bottles specially labeled for them and they were warned not to exceed these levels. Immediately after the training session, BM was measured in all conditions and then the shooting performance test was administered again and the session ended. A rest period of at least 48h was given between the sessions.

### 2.2. Participants

The study included 47 healthy adolescent male handball players aged between 13 and 18 years (15.00 ± 1.49 years). The participants engaged in handball training no less than three days a week, had an average of at least one year of training experience, and had not suffered any upper extremity injuries in the previous six months. The exclusion criteria included having an injury in the last six months [19]. G-Power 3.1.9.4 software (Germany) was used to calculate the number of participants included in the study and determine the required sample size. The sample size was determined as 45 participants based on an effect size of f = 0.5, an alpha (error) rate of 5%, and a power of 95% [17,19], and 47 participants were included by anticipating potential risks.

#### Determination of Participant Groups

The participants included in the study [n = 47] were divided into performance groups according to the shooting performance test in the familiarization session. This distinction was made by mean-split analysis based on the mean values (4.89) and SD (±1.8) of the shooting accuracy of all participants who performed the test [20]. Participants below the mean (0–3 hits/10 shots) formed the Low Performer (LP) group [n = 17]; participants near the mean (4–7 hits/10 shots) formed the Average Performer (AP) group [n = 26]; and participants above the mean (7–10 hits/10 shots) formed the High Performer (HP) group [n = 8]. All participants and their families were informed about the study and completed the informed consent form and their voluntary participation in the study was documented. The study was approved by Manisa Celal Bayar University Health Sciences Ethics Committee with a decision dated 7 June 2023, number 20.487.486/1870. The Declaration of Helsinki guided the conduct of this study.

### 2.3. Pre-Session Standardization

Participants were subjected to fluid and food restrictions 2 h before each session. Hydration level was monitored using a subjective method and body mass change because the study contains a precise and standardized temporal transition phase between the pre-shooting performance test—handball training—and post-shooting performance test, and the reflection of fluid intake on biomarkers (urine or blood) is time dependent [21] and individualistic [22]. At the beginning of each session, participants’ subjective hydration levels were assessed using an analogue scale asking about thirst intensity, with categories ranging from 1 (‘not thirsty at all’) to 3 (‘extremely thirsty’) [23]. Standardization was achieved by fully hydrating before the conditions [24,25].

### 2.4. Familiarization Session

In the first stage, demographic information was collected after the experimental protocol of the study was explained. Following the anthropometric measurements, the shooting performance test was explained to the participants. After the explanation, the teaching and application of the shooting performance test was carried out by making 10 trial shots to be used in the test. In this section, the aim was to familiarize the participants with the shooting performance protocol and to reduce possible performance variability in the initial measurements.

### 2.5. Anthropometric Measurements

The height of the participants was measured by the same person using a tape measure, standing upright barefoot with their backs against a wall, determining the top of their heads, and recorded in centimeters. BM was measured using a body analysis scale (Arzum Fitsense, Model: AR 55, Istanbul, Turkey) electronic weighing scale and recorded in kilograms (accuracy 0.1 kg) [26,27].

### 2.6. Measurement of Hydration Status

The BM of the participants was measured before and after the training sessions, and the formula (post-BM–pre-BM/pre-BM) was used to calculate the degree of dehydration [28,29].

### 2.7. Shooting Performance Test

The chosen shooting performance test is widely validated in handball performance research, allowing for both inter- and intra-subject reliability due to its standardized distance and timing controls [30,31,32]. In the test (Figure 1), the participants have 10 throws from behind the 7 m line to a 1 m × 1 m plastic plate hung in the middle of the handball goal. The participants performed the test with the highest speed and accuracy they could with their dominant hand. The time between the throws was set as 5 s for the participant to maintain attention and concentration [33]. During the throws, the speed of the handball leaving the hand and traveling to the goal was measured with a branded radar device, a Bushnell radar gun, and a recognized gold-standard radar gun (JUNGS, Model: 101911, CA, USA), which can measure speeds between 10 and 331 km/h and has an accuracy of ±0.8 km/h [34]. The measurements obtained were recorded in km/hour [35,36].

### 2.8. Handball Training Programme

Participants were subjected to a 60 min session of low- and medium-intensity actions designed by an expert coach. The session included:-Warm-up (10 min): Low-to-moderate running and basic passing technique drills + dynamic stretching exercises;-Main phase (40 min): Technical passing, positional shots, and small-sided games with positional differences;-Cool down (10 min): 5 min running + 5 min stretching exercises were carried out.

In the main phase, passing and shooting techniques and tactical variations frequently used in the match were preferred.

#### Water and Sports Drink Intake Process

Participants received water and sports drink supplements at 5 min intervals (1% × body mass = 12 equal parts) during handball training [3]. In the water intake condition, participants received ‘Natural Water’, commercial water [37]. The sodium and carbohydrate content in the sports drink was selected to both replenish electrolytes and provide an immediate energy source, as recommended in intermittent sports contexts. The sports drink intake session included an ‘Isotonic Sports Drink’, a commercial sports drink (Powerade) that is rich in water, carbohydrates, and electrolytes. This drink contains 170 kcal/L, 39 g/L carbohydrate (sugar), 0 g fat, 0 g protein, and 1.2 g/L sodium [3,38]. The commercial water and sports drink temperature was 10–12 °C [39].

### 2.9. Statistical Analysis

All statistical analyses were performed with the software package IBM- SPSS (version 23.01). The Shapiro–Wilk test was applied to determine the normal distribution of all participants’ data (BM, Body Mass Index; (BMI), shooting accuracy, and shooting speed). It was determined that the data were normally distributed. A three-way mixed ANOVA was conducted to investigate the effects of the type of drink (fluid restriction, water intake, and sports drink intake) on accuracy scores and speed changes depending on the group (Low Performers, Average Performers, and High Performers) in terms of time (pre and post). Using Partial Eta Squared (η_p_^2^) and F values determined effect sizes between the data. The significance level was set as less than 0.05 [40].

## 3. Results

### 3.1. Descriptive Characteristics of the Participants

The demographic information and shooting performance of the 47 participants can be seen in Table 1. Descriptive statistics indicated no significant differences across groups for age, training experience, height, or body mass, confirming comparable baseline characteristics.

### 3.2. Hydration Levels

Three-way ANOVA was performed for the changes in body mass with fluid intake (water intake and sports drink intake) and fluid restriction in the performance groups (Low Performers, Average Performers, and High Performers). The main effects of factors and interaction were evaluated. There was no significant main effect of fluid intake, F(2.88) = 047, *p* = 0.955, η_p_^2^ = 0.001. Body mass loss in the Average Performer (Mean (M) = 63.01 kg, SD (Standard Deviation) = 3.76 kg) group was significantly higher than in the Low Performer (M = 62.00 kg, SD = 3.90 kg) and High Performer (M = 62.00 kg, SD = 3.90 kg) groups (*p* = 0.37). Post hoc analyses revealed that only in the Average Performer group was body mass significantly decreased (*p* = 0.000) with fluid restriction (M = 62.05 kg, SD = 3.2 kg) compared to water intake (M = 63.41 kg, SD = 3.06 kg). Figure 2 displays the notation associated with the change processes (see Figure 2).

### 3.3. Shooting Accuracy

Three-way ANOVA analysis was performed to evaluate the main effects of fluid intake (water intake and sports drink intake) and fluid restriction on shooting accuracy changes in the performance groups (Low Performers, Average Performers, and High Performers), and the main effects of the factors and their interaction were evaluated (see Figure 3). There was a significant main effect of the fluid intake process on shooting accuracy change, F(2, 88) = 34.32, *p* < 0.001, η_p_^2^ = 0.44. In post hoc analyses, it was found that water intake (M = 4.77, SD = 0.24) and sports drink (M = 4.93, SD = 0.22) intake (M = 356, SD = 0.15) increased shooting accuracy values (*p* < 0.001) compared to fluid restriction (M = 2.86, SD = 0.24).

There was a statistically significant effect in the three-way interaction (type of fluid intake × times × performance level), F(2, 57) = 2.34, *p* = 0.10, η_p_^2^ = 0.332. In post hoc analyses, it was found that sports drink intake influenced the Low Performers (M = 3.53, SD = 0.49), Average Performers (M = 5.65, SD = 0.35), and High Performers (M = 7.00, SD = 0.63) in achieving higher shooting accuracy values compared to the Low Performer group (*p* < 0.001).

#### Considering Group-Specific Changes at Shooting Accuracy

The Low Performer group had increased shooting accuracy performance after water intake (M = 4.00, SD = 0.55) and sports drink intake (M = 3.53, SD = 0.49) compared to the fluid restriction (M = 0.61, SD = 0.14) condition, with the highest effect occurring in the water intake condition (*p* < 0.001).

In the Average Performer group, shooting accuracy performance increased after water intake (M = 5.19, SD = 0.38) and sports drink intake (M = 5.65, SD = 0.35) compared to the fluid restriction (M = 0.53, SD = 0.10) condition, with the highest effect occurring in the sports drink intake condition (*p* < 0.001).

In the High Performer group, shooting accuracy performance increased after water intake (M = 4.87, SD = 0.70) and sports drink intake (M = 7.00, SD = 0.63) compared to the fluid restriction (M = 0.62, SD = 0.18) condition, with the highest effect occurring in the sports drink intake condition (*p* < 0.001).

Planned contrasts support the present results by revealing that any fluid intake (water intake and sports drink intake) significantly increased shooting accuracy values compared to fluid restriction F(1, 44) = 87.06, *p* < 0.001, η_p_^2^ = 0.664. Figure 3 displays the notation associated with the change processes.

### 3.4. Shooting Speed

Three-way ANOVA analyses were performed to evaluate the main effects of the factors and the interaction with the shooting speed changes of fluid intake (water and sports drink intake) and fluid restriction according to the performance groups (Low Performers, Average Performers, and High Performers). There is a significant main effect of the fluid intake process on shooting speed change, F(1, 61) = 4.05, *p* = 0.35, η_p_^2^ = 0.84 (see Figure 4).

There was a significant effect of fluid intake x time interaction, F(1, 29) = 14.38, *p* < 0.001, η_p_^2^ = 0.246. Post hoc analyses revealed that fluid intake (M = 48.72 km/h, SD = 1.37) increased shooting speed compared to fluid restriction (M = 46.36 km/h, SD = 1.68), and this effect was most pronounced in the water intake condition (*p* = 0.038).

#### Considering Group-Specific Changes at Shooting Speed

The Low Performer group’s shooting speed increased, but not significantly (*p* = 0.141), after water intake (45.19 km/h, SD = 2.33) and sports drink intake (M = 45.01 km/h, SD = 2.34) compared to the fluid restriction condition (M = 42.46 km/h, SD = 2.85).

The Average Performer group’s shooting speed was increased after water intake (M = 47.68 km/h, SD = 1.65) and sports drink intake (M = 48.11 km/h, SD = 1.66) compared to the fluid restriction condition (M = 45.84 km/h, SD = 2.02), with the greatest effect occurring in the sports drink intake condition (*p* < 0.011). 

In the High Performer group, there was a decrease in the shooting speed value in the fluid restriction condition (M = 50.75 km/h, SD = 3.64) compared to the water intake condition (M = 53.30 km/h, SD = 2.97) (*p* < 0.001). In this group, speed increased after water and sports drink intake conditions. 

Planned contrasts support the present results by revealing that any fluid intake (water intake and sports drink intake) significantly increased shooting speed values compared to fluid restriction F(1, 44) = 4.29, *p*: 0.044, η_p_^2^ = 0.89.

## 4. Discussion

This study aimed to investigate the changes in shooting performance and speed of adolescent handball players under fluid restriction and different fluid intake conditions. The main findings of the study were as follows: (i) fluid restriction during 60 min of handball training negatively affected shooting accuracy and speed in adolescent handball players associated with hypohydration, (ii) water and sports drink intake positively affected shooting accuracy and speed, and (iii) fluid restriction and fluid intake protocols affect shooting accuracy and speed at all shooting performance levels.

Fluid intake compared to fluid restriction produced a positive main effect on post-session values of shooting accuracy, F(2, 88) = 34.32, *p* < 0.001, η_p_^2^ = 0.44, and shooting speed, F(1, 61) = 4.05, *p* = 0.35, η_p_^2^ = 0.84. In a literature study conducted on sub-elite male basketball players, shooting accuracy levels were evaluated under fluid restriction and water intake conditions for a 40 min moderate-intensity training period; no difference in shooting accuracy performance was observed between the conditions [4]. Devlin et al. evaluated shooting performance in sub-elite cricketers under fluid restriction and fluid intake conditions induced by ~1 h intermittent exercise. As a result, they reported gradual deterioration in shooting accuracy in the fluid restriction condition [7]. Focusing on sub-elite basketball players, Baker et al. investigated the effects of fluid restriction on shooting performance in basketball. In the study, the athletes performed a 3 h intermittent walking exercise on a treadmill followed by free shooting from the foul line. As a result of the study, they found that the participants’ shooting accuracy rate deteriorated under the hypohydration condition [8]. The results of our study coincide with the information reported in the literature regarding adult athletes. In particular, the process we followed in common with the above studies is that fluid restriction has a detrimental effect on shooting accuracy and speed. This suggests that the participants experienced additional stress due to fluid restriction in addition to the stress they were already experiencing due to exercise. In addition, adolescent athletes are known to excessively breathe and sweat due to their metabolic levels. This may affect the rate of fluid loss more negatively. Therefore, handball training caused 1% hypohydration in all participants. Hypohydration is known to cause cognitive and motor coordination problems. Shooting accuracy and speed are closely related to cognitive and motor characteristics such as reaction time, visuomotor tracking, timing, and intermuscular coordination [1]. Hypohydration has been shown to impair neuromuscular coordination, which is integral to accuracy in shooting sports, especially under the cognitive load imposed by the fast-paced demands of handball [41]. In our study, hypohydration may have affected the shooting performance of adolescent athletes with little training experience by impairing communication between the parts. Repeated shots in the throwing test protocol also more clearly revealed the potential for badly conditioned adolescent handball players to be affected by restriction. With this information, it was thought that higher-order associations could explain the effects of this impairment.

According to the group analysis, the Low Performer group’s shooting accuracy improved after drinking water and sports drinks compared to when they had their fluids restricted. The effect was strongest with water intake (*p* < 0.001). When compared to the fluid restriction condition, the shooting speed values increased numerically (*p* = 0.141) but not significantly. The Average Performer group showed improved shooting accuracy after water intake and sports drink intake compared to the fluid restriction condition. The greatest improvement occurred with sports drink intake (*p* < 0.001). In addition, shooting speed values increased after water intake and sports drink intake compared to fluid restriction, and the greatest effect occurred after sports drink intake (*p* < 0.011). The High Performer group’s shooting accuracy improved after water intake and sports drink intake compared to when they had their fluids restricted. The greatest improvement occurred when they consumed sports drinks (*p* < 0.001). On the other hand, shooting speed values improved after drinking water and sports drinks compared to when they had their fluids restricted (*p* < 0.001). The literature was examined, and it was reported that water intake (as much as the BM lost) after each period of a 40 min low-intensity basketball competition with national-level female basketball players improved the number of shootings by 5.3 ± 2.8% and the overall shooting accuracy rate by 30–35% [42]. In Burke et al.’s study, a 2 h simulated tennis competition was performed with tennis athletes. During the experiment, one group had a regular water intake of 505 milliliters, while the other group was subjected to fluid restriction. After the competition, a shooting performance test consisting of 50 shots was applied. As a result, the hit rate in the water intake condition was 49.45%, while the rate in the fluid restriction condition was 48.65% [9,43]. In adolescent athletes, electrolyte losses, especially sodium, occur at a high sweating rate independent of environmental conditions (temperature and humidity). The potential of these losses to decrease performance is known [44]. In the handball training protocol applied in our study, intermittent water intake equal to BM loss had a positive effect on shooting performance. The significant improvement in all of the groups suggests that shooting performance parameters are positively affected by water intake in this training protocol, possibly slowing the deterioration in fluid–electrolyte balance.

The limited literature on sports drink intake and shooting performance was examined. Carvalho et al. examined the effects of 90 min training sessions performed under conditions of fluid loss, water intake (3.8 mg/L sodium), and sports drink intake (7.2% sugar, 0.8% maltodextrin, and 510 mg/L sodium) on basketball shooting (2- and 3-point shootings and free shooting) performance [3]. As a result of the study, it was reported that the 2-point (55.0 ± 20.2%), 3-point (36.7 ± 19.7%), and free shooting (59.2 ± 22.8%) performance of adolescent basketball players (14.8 ± 0.45 years) was impaired in the fluid restriction condition. In the water intake condition, improvements were observed in 2-point (60.8 ± 12.4%), 3-point (37.5 ± 16.0%), and free shooting (62.5 ± 20.1%) values. However, the greatest improvement in 2-point (60.0 ± 16.5%), 3-point (42.5 ± 16.0%), and free shooting (65.8 ± 20.2%) values was achieved in the sports drink intake condition [45]. In another study, adolescent basketball players were subjected to a fluid restriction condition and given a sports drink with 6% carbohydrate content in the other condition during a 2 h intermittent exercise session. As a result, while the free shooting accuracy percentage was 60 ± 8% in the sports drink intake condition, this rate decreased to 45 ± 9% in the fluid restriction condition. Baker et al. performed a study involving a 3 h intermittent treadmill exercise in sub-elite basketball players (17 ± 28 years) under fluid restriction and sports drink (0% carbohydrate and 18% sodium) intake conditions. Free shooting performance was evaluated immediately after the exercise [46]. As a result of the study, while the accuracy rate was 86 ± 6% in the sports drink intake condition, this rate was found to be 83 ± 3% in the fluid restriction condition. Considering the information presented, it can be seen that the positive results we obtained are highly compatible with the literature.

To the best of our knowledge, this is the first study specific to adolescent handball players conducted with this research design. The carbohydrates and electrolytes in the sports drink may have a positive effect on handball shooting accuracy and shooting speed by improving the mechanisms related to cognitive and motor coordination, thus indirectly increasing shooting performance. Sports drink supplementation in an intermittent model in adolescent handball players has a high potential to positively affect shooting performance parameters.

According to our planned contrast analysis performed to evaluate the results from a general perspective, it was found that fluid intake (water intake and sports drink intake) significantly increased the shooting accuracy values compared to fluid restriction F(2, 57) = 2.34, *p* = 0.10, η_p_^2^ = 0.332. In addition, it was also found that fluid intake (water intake and sports drink intake) significantly increased shooting speed values compared to fluid restriction, F(1, 44) = 4.29, *p*: 0.044, η_p_^2^ = 0.89, and these findings were supported by the literature findings presented above. In both conditions (water intake and sports drink intake), shooting accuracy and speed increased compared to fluid restriction. These results indicate that the shooting performance of male adolescent handball players is so sensitive that it can be affected by even low levels of hypohydration, whereas fluid intake can be considered as an indicator that fluid intake is critical for shooting performance.

## 5. Limitations

Performance fluctuations in shooting accuracy and speed values occurred in the participants in all sessions. Previous research has shown that fluctuations in accuracy and speed values in handball shooting performance are affected by training experience [33]. The fact that possible accuracy decreases that may occur due to the increase in shooting speed was not observed in our study eliminates the possibility that speed fluctuations are caused by focusing on accuracy. On the other hand, the subjective hydration questionnaire we used in this study, although practical, may be prone to bias and lack specificity. Future studies may consider including measures of urine osmolality or specific gravity to objectively monitor hydration status. They could discuss possible confounding variables such as individual variability in sweat rates or previous hydration levels; uncontrolled variables such as individual sweat rates and previous hydration status may introduce variability into the results. A standardized pre-test hydration protocol or individual sweat rate monitoring may reduce the impact of these factors in future research. In this study, fluid deficit was calculated based on the body weight of the participants before the experimental design. One of the limitations of this study is that the electrolyte balance of the participants could not be monitored and determined during the period before the start of the experiment (i.e., the last week). In future studies, it is important to monitor fluid intake for a longer period before the experimental procedure. Finally, since our research was conducted in a controlled indoor environment and only male adolescents, the findings may not be fully generalized to outdoor conditions (different temperatures and humidities) and female adolescents.

## 6. Conclusions

The findings of this study reveal that shooting accuracy and speed parameters in adolescent handball players are sensitive enough to be negatively affected by low-level hypohydration, while fluid intake, on the contrary, improved shooting accuracy and speed. It was also observed that sports drink intake in particular had an ergogenic effect on shooting performance. Therefore, these findings revealed that hydration is a key factor in the shooting performance of adolescent handball players. 

Based on the findings, adolescent handball players should be strongly encouraged to consume fluids during training and matches, especially during periods of high intensity, to maintain shooting performance.

It is also recommended that handball field practitioners, nutritionists, coaches, and athletes utilize the fluid intake protocols implemented in this study to maintain and improve shooting performance. 

## Figures and Tables

**Figure 1 nutrients-16-04246-f001:**
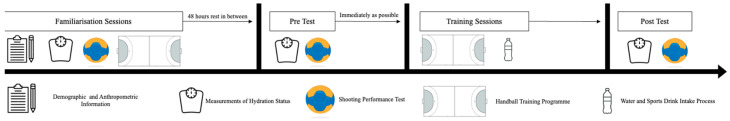
The design of the study.

**Figure 2 nutrients-16-04246-f002:**
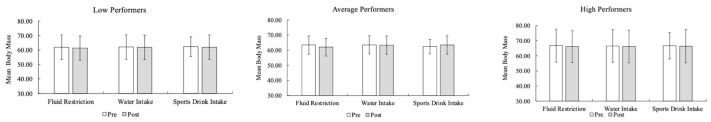
Three-way ANOVA results of the body mass change between performance groups and fluid intake conditions. The **left panel** (LP), **average panel** (AP), and **right panel** (HP) groups are shown. The light-colored column represents pre-test values, and the dark-colored column represents post-test values. The vertical axis represents body mass values.

**Figure 3 nutrients-16-04246-f003:**
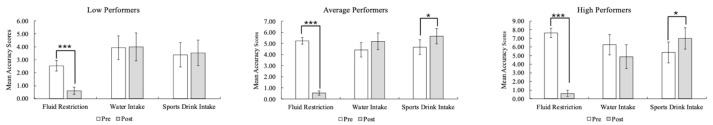
Three-way ANOVA results of the shooting accuracy change between performance groups and fluid intake conditions. The **left panel** (LP), **average panel** (AP), and **right panel** (HP) groups are shown. The light-colored column represents pre-test values, and the dark-colored column represents post-test values. The vertical axis represents body mass values. * *p* < 0.05, *** *p* < 0.001.

**Figure 4 nutrients-16-04246-f004:**
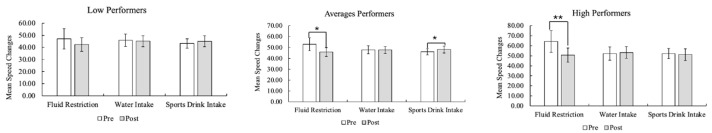
Three-way ANOVA results of the shooting speed change between the performance groups and fluid intake conditions. The **left panel** (LP), **average panel** (AP), and **right panel** (HP) groups are shown. The light-colored column represents pre-test values, and the dark-colored column represents post-test values. The vertical axis represents body mass values. * *p* < 0.05, ** *p* < 0.01.

**Table 1 nutrients-16-04246-t001:** Descriptive characteristics of the participants.

Group	Age (Years)	Training Age (Years)	Height (cm)	Body Mass (kg)	BMI (kg/m^2^)	Shooting Accuracy (0–10)	Shooting Speed (km/Average)
LP (n = 13)	15 ± 1.35	1.31 ± 0.48	171.85 ± 8.26	62.11 ± 9.07	21.05 ± 2.99	2.54 ± 0.66	47.06 ± 7.80
AP (n = 26)	15.07 ± 1.59	1.42 ± 0.50	171.88 ± 9.90	63.44 ± 15.54	21.28 ± 3.91	5.23 ± 0.76	53.02 ± 14.88
HP (n = 8)	14.75 ± 1.58	1.88 ± 0.64	173.63 ± 16.22	66.8 ± 23.53	21.475 ± 4.58	7.63 ± 0.92	64.39 ± 25.14

Note: LP: Low Performer, AP: Average Performer, HP: High Performer, and BMI = Body Mass Index.

## Data Availability

All collected data in the current study are available after obtaining permission from all of the authors. Written proposals can be addressed to the corresponding authors for appropriateness of use. The data are not publicly available due to privacy and ethical reasons.

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
