# Peer review of "The Effect of Fluid Restriction and Intake Conditions on the Shooting Performance of Competitive Adolescent Handball Players"

_nutrients, 2024, doi:10.3390/nu16234246_

Round 1
Reviewer 1 Report
Comments and Suggestions for Authors
The study is interesting but needs major changes.
Introduction
- Line 44. Repeated reference
- Line 47. Please specify how it affects performance. Please specify more on metabolic and neuromuscular level. If possible refer to the book: Heat Stress in Sport and Exercise.
- Line 55. You seem to have missed some words
- Line 57. Remove the double space
- Line 60. Specify whether these sports drinks contained taurine, caffeine, guarana....
- The introduction should describe more why dehydration can affect marksmanship, how it negatively affects motor as well as cognitive-motor skills.
Methods
- Please describe the warm-up more
- Why was a standard handball warm-up not applied? There are many muscles involved in throwing handball, moreover, if there was a training session afterwards it would have been applied for both.
- Why 1% intake if dehydration in training can be higher, even more so at the temperatures that were applied. Indicate according to previous studies what is dehydration in a handball training session.
- Was there a nutritional survey of the participants to find out the electrolytes consumed during the week of measurement? This is very important and not having done so is a major limitation.
- Line 109, please keep the text in the same format as the rest of the text.
- Line 144. Please correct ‘thne’.
- It is not clear how the experiment was organised. Was there one or two groups? Were participants randomised? Was it a crossover study?
- In the statistical analysis an ANOVA should be performed. If it was the same group, it should be a repeated variables ANOVA. On the other hand, if it was different groups, a two-way ANOVA should be performed. Please also include the effect size.
Discussion
Well established.
Introduction
- Line 44. Repeated reference
- Line 47. Please specify how it affects performance. Please specify more on metabolic and neuromuscular level. If possible refer to the book: Heat Stress in Sport and Exercise.
- Line 55. You seem to have missed some words
- Line 57. Remove the double space
- Line 60. Specify whether these sports drinks contained taurine, caffeine, guarana....
- The introduction should describe more why dehydration can affect marksmanship, how it negatively affects motor as well as cognitive-motor skills.
Methods
- Please describe the warm-up more
- Why was a standard handball warm-up not applied? There are many muscles involved in throwing handball, moreover, if there was a training session afterwards it would have been applied for both.
- Why 1% intake if dehydration in training can be higher, even more so at the temperatures that were applied.
- Was there a nutritional survey of the participants to find out the electrolytes consumed during the week of measurement? This is very important and not having done so is a major limitation.
- Line 109, please keep the text in the same format as the rest of the text.
- Line 144. Please correct ‘thne’.
- It is not clear how the experiment was organised. Was there one or two groups? Were participants randomised? Was it a crossover study?
- In the statistical analysis an ANOVA should be performed. If it was the same group, it should be a repeated variables ANOVA. On the other hand, if it was different groups, a two-way ANOVA should be performed. Please also include the effect size.
Discussion
Well established.
Author Response
Reviewer 1
The study is interesting but needs major changes.
INTRODUCTION
- Line 44. Repeated reference
Author's Response: We appreciate the reviewer’s comment. Reference repetition has been corrected.
[8]
- Line 47. Please specify how it affects performance. Please specify more on metabolic and neuromuscular level. If possible refer to the book: Heat Stress in Sport and Exercise.
Author's Response: We thank the reviewers’ contribution. The specific effects of hypohydration have been added with reference.
Hypohydration (-1 to 2% BM) induced by fluid restriction significantly negatively affects cognitive-motor task performance. This impairs motor control and thus worsens shooting performance [54].
- Line 55. You seem to have missed some words
Author's Response: We appreciate the reviewer’s comment. The sentences containing the words that disrupt the flow of the sentence are organised according to the flow of meaning.
Adolescents have smaller body surface area-to-mass ratios and variable sweat rates than adults, which requires specialised hydration approaches, especially in high-intensity sports such as handball [1].
- Line 57. Remove the double space
Author's Response: We appreciate the reviewer’s comment. In line 57, the double space has been removed.
- Line 60. Specify whether these sports drinks contained taurine, caffeine, guarana....
Author's Response: We thank the reviewer’s comment. Sports drinks containing carbohydrates have been specified and corrected.
Carvalho et al. in their study on fluid restriction in adolescent basketball players, showed that shot accuracy and velocity did not change under hypohydration conditions, but performance improved with the consumption of sports drinks (carbohydrate solution) [3]. In their review, Barnes et al. reported that carbohydrate-containing sports drinks consumed during training sessions indirectly increased shooting accuracy at the end of training [12].
- The introduction should describe more why dehydration can affect marksmanship, how it negatively affects motor as well as cognitive-motor skills.
- Author's Response: We thank the reviewers’ contribution,
Hypohydration (-1 to 2% BM) induced by fluid restriction significantly negatively affects cognitive-motor task performance. This impairs motor control and thus worsens shooting performance [54].
METHODS
- Please describe the warm-up more. Why was a standard handball warm-up not applied? There are many muscles involved in throwing handball, moreover, if there was a training session afterwards it would have been applied for both.
In the warm-up period (total ten minutes), low to moderate running, and basic passing technique drills followed by dynamic stretching exercises with upper extremity intensity were performed. Then, the shooting performance test was started. Afterwards, the participants were shooting to make 10 shots on the plate hung on the handball goal, while remaining stationary on a 7 m handball line (Figure 1-b).
- - Why 1% intake if dehydration in training can be higher, even more so at the temperatures that were applied. Indicate according to previous studies what is dehydration in a handball training session.
Author's Response: We thank the reviewers’ contribution. The text has been amended as follows;
This rate was calculated by using the body mass loss rate per 60 minutes of handball training during the familiarisation session and the study of Rodrigez et al. [55].
- Was there a nutritional survey of the participants to find out the electrolytes consumed during the week of measurement? This is very important and not having done so is a major limitation.
Author's Response: We thank the reviewers’ contribution. Similarity in hydration levels was determined using body weight measurement before the participants entered the experiment. Determination of fluid-electrolyte balance by nutritional monitoring could have had a supportive effect in this study. This is stated in the limitations section.
In this study, fluid deficit was calculated based on the body weight of the participants before the experiment design. One of the limitations of this study is that the electrolyte balance of the participants could not be monitored and determined during the period before the start of the experiment (i.e. the last week).
- Line 109, please keep the text in the same format as the rest of the text.
Author's Response: We thank the reviewers’ attention. Thank you for the interest of the critics. The relevant section has been redesigned in the same format.
- Line 144. Please correct ‘thne’.
- Author's Response: We thank the reviewer’s comment. The relationship contex
In the test (Figure 1b) then, participants have ten throws from behind the 7-m.
- It is not clear how the experiment was organised. Was there one or two groups? Were participants randomised? Was it a crossover study?
Author's Response: We thank for expanding and writing up the limited information that a single study group, included in the study according to their performance, participated in all sessions and their performance in these sessions was evaluated. In addition, the details on participant selection and distribution have been defined by adding the heading ‘Determination of Participant Groups’.
The study was conducted in four sessions (Figure 1-a) including one familiarisation session and three experimental intervention (fluid restriction, water and sports drink intake) conditions. A single research group was included in each of these three conditions. Participants who were divided into low-average-high performance groups according to the shooting performance groups in the famillarisation condition were included in the fluid restriction, water, and sports drink groups, respectively. The study was planned with a randomised crossover design so that the participants were evaluated in all conditions.
Determination of Participant Groups
The participants included in the study [N = 47] were divided into performance groups according to the shooting performance test in the famillarisation session. This distinction was made by mean-split analysis based on the mean values (4.89) and SD (±1.8) of the shooting accuracy of all participants who performed the test [56]. Participants below the mean (0-3 hits/10 shots) formed the Low Performer (LP) group [n = 17]; participants near to the mean (4-7 hits/10 shots) formed the Average Performer (AP) group [n = 26]; and participants above the mean (7-10 hits/10 shots) formed the High Performer (HP) group [n = 8].
- In the statistical analysis an ANOVA should be performed. If it was the same group, it should be a repeated variables ANOVA. On the other hand, if it was different groups, a two-way ANOVA should be performed. Please also include the effect size.
Author's Response: First, our statistical design error is deeply regrettable. In line with your valuable suggestion, we reapplied ‘Three-way Anova’ analysis to our groups determined by ‘the ‘mean-split’ method. Your valuable suggestions have significantly contributed to the development of our article and strengthened our findings.
Due to the development of statistical results resulting from the changed analyses, the discussion and conclusion sections have also been improved and the relevant parts have been coloured in green.
We have addressed the shortcomings in our group separation method by incorporating your valuable suggestions. You can find this section under the title ‘Determination of Participant Groups’ between lines 113-120 of the manuscript.
*Lines 183–192 under the heading '2.10. Statistical Analysis' provide method information about the three-way ANOVA analysis.
*You can find our findings updated as a result of the analysis between lines 202-281.
*Discussion sections updated in line with the findings are between lines 286-292, 320-332, and 374-384.
2.10. Statistical Analysis
All statistical analyses were performed with the software package IBM- SPSS (version 23.01). The Shapiro-Wilk test was applied to determine the normal distribution of all participants' data (Body Mass), Shooting accuracy, and Shooting speed). It was determined that the data were normally distributed. A three-way mixed ANOVA was conducted to investigate the effects of the type of drink (Fluid Restriction, Water Intake, Sports Drink Intake) on accuracy scores and speed changes depending on the group (Low Performers, Average Performers, High Performers) in terms of time (Pre & Post). Statistical significance F values and effect sizes Using Partial Eta Squared (ηp2) between the data. The significance level was set as less than 0.05 [57].
Reviewer 2
The purpose of the study is to address the effect of hydration on shooting performance in adolescent handball players under three conditions: fluid restriction, water intake, and sports drink consumption. Taking into account the physiological and metabolic distinctions between adults and adolescents, the study clarifies the special effects of hydration on male adolescents, a vulnerable but essential group in sports.
The topic is interesting and pertinent to the knowledge of obesity and lifestyle choices. The study simulates actual match conditions while upholding scientific rigor by using a well-controlled setting with constant humidity and temperature. Using validated methods such as the JUGS Gun to measure shooting velocity and accuracy guarantees quantifiable and repeatable results.
Accuracy and shooting speed and are essential performance metrics in handball. Training plans and match tactics can be immediately enhanced by knowing how hydration affects these measures.
Although the study is clearly written, well-organized, and uses an easy-to-follow terminology there are certain places where the methodology might be broadened or enhanced for increased reliability and validity.
I have some comments/suggestions in order to improve quality of the study:
INTRODUCTION
- The Abstract is too long. The authors should respect the requirements specified in the instructions for the authors of Nutrients journal (https://www.mdpi.com/journal/nutrients/instructionsconcerning the maximum length of 200 words for the Abstract (currently it has 220 words);
Author's Response: We thank the reviewer’s for the comment. The abstract was reorganized according to the 200-word criterion determined by the journal in line with the reviewer's suggestion.
- In the Introduction, the purpose of the work and its significance are defined and the three specific hypotheses being tested are included. The authors should clearly specify why they consider that their research is original and relevant to the field and what is added to the subject area compared with other cited materials;
- Author's Response: We thank the reviewer’s contribution, We apologise for the ambiguity. With the reviewer’s suggestion, our aims and hypotheses have been reorganized more understandably.
The shooting accuracy and speed are effects of the handball game results. Therefore, it is important to investigate the parameters (accuracy and speed) affecting shooting performance. For these reasons, this study aimed to investigate the effects of fluid restriction and different types of fluid intake on shooting performance and shooting speed in adolescent handball players.
Main hypotheses of this study; i) fluid restriction during training may impair shot accuracy and speed in adolescent handball players, ii) water and sports drink intake may positively affect shot accuracy and speed, iii) fluid restriction and fluid intake protocols may affect shot accuracy and speed at different levels according to shot performance level.
- The acronyms BM, LR are not defined into the text;
Author's Response: We thank the reviewer’s contribution. In line 27, the BM abbreviations are explained. Line 191 LR abbreviations are excluded.
Body Mass (BM)
- The Note from lines 190-191 seems not finished;
Author's Response: We thank the reviewer’s contribution, In lines 190 and 191 reorganized.
- In line 338, a reference has to be included for” Baker and al.”
Author's Response: We thank the reviewer’s. At line 338, a reference number for ‘Baker et al’ has been added.
Baker et al., sub-elite basketball players investigated the effects of fluid restriction on shooting performance in basketball. In the study, the athletes performed 3-h. intermittent walking exercise on a treadmill followed by free shootings from the foul line. As a result of the study, they found that the participants's shooting accuracy rate deteriorated under the hypohydration condition [8].
- The study should include as limitationsthe following aspects:
- Only male teenage handball players were included in the sample, which limited its applicability to female players. The findings would be more broadly applicable if the authors extended their study to female athletes in the future and looked into any potential variations in the effects of hydration caused by physiological variables, such as hormonal changes, that alter fluid balance;
- The study looks at performance in brief sessions without addressing the long-term effects of hydration practices on skill development or recovery;
Author's Response: Thank you for your valuable contribution on limitations. Limitations related to gender difference, female athletes and prolonged hydration processes have been added to the relevant section.
In this study, fluid deficit was calculated based on the body weight of the participants before the experiment design. One of the limitations of this study is that the fluid intake of the participants could not be monitored and determined during the period before the start of the experiment (i.e. the last week). In future studies, it is important to monitor fluid intake for a longer period before the experimental procedure. Finally, since our research was conducted in a controlled indoor environment and only male adolescents, the findings may not be fully generalised to outdoor conditions (different temperature and humidity ) and female adolescents.
- The description of the references should respect the requirements specified in the instructions for the authors of Nutrients journal (https://www.mdpi.com/journal/nutrients/instructions). The authors should respects this requirement.
- Author's Response: We thank the reviewer’s comment. References are organised in accordance with the rules of the journal.

Reviewer 2 Report
Comments and Suggestions for Authors
The purpose of the study is to address the effect of hydration on shooting performance in adolescent handball players under three conditions: fluid restriction, water intake, and sports drink consumption. Taking into account the physiological and metabolic distinctions between adults and adolescents, the study clarifies the special effects of hydration on male adolescents, a vulnerable but essential group in sports.
The topic is interesting and pertinent to the knowledge of obesity and lifestyle choices. The study simulates actual match conditions while upholding scientific rigor by using a well-controlled setting with constant humidity and temperature. Using validated methods such as the JUGS Gun to measure shooting velocity and accuracy guarantees quantifiable and repeatable results.
Accuracy and shooting speed and are essential performance metrics in handball. Training plans and match tactics can be immediately enhanced by knowing how hydration affects these measures.
Although the study is clearly written, well-organized, and uses an easy-to-follow terminology there are certain places where the methodology might be broadened or enhanced for increased reliability and validity.
I have some comments/suggestions in order to improve quality of the study:
1. The Abstract is too long. The authors should respect the requirements specified in the instructions for the authors of Nutrients journal (https://www.mdpi.com/journal/nutrients/instructionsconcerning the maximum length of 200 words for the Abstract (currently it has 220 words);
2. In the Introduction, the purpose of the work and its significance are defined and the three specific hypotheses being tested are included. The authors should clearly specify why they consider that their research is original and relevant to the field and what is added to the subject area compared with other cited materials;
3. The acronyms BM, LR are not defined into the text;
4. The Note from lines 190-191 seems not finished;
5. In line 338, a reference has to be included for ”Baker and al.”
6. The study should include as limitations the following aspects:
• Only male teenage handball players were included in the sample, which limited its applicability to female players. The findings would be more broadly applicable if the authors extended their study to female athletes in the future and looked into any potential variations in the effects of hydration caused by physiological variables, such as hormonal changes, that alter fluid balance;
• The study looks at performance in brief sessions without addressing the long-term effects of hydration practices on skill development or recovery;
7. The description of the references should respect the requirements specified in the instructions for the authors of Nutrients journal (https://www.mdpi.com/journal/nutrients/instructions). The authors should respects this requirement.
I hope my feedback is useful to the author in improving their paper and wish him all the best in pursuing this important area of research.
Author Response
Reviewer 2
The purpose of the study is to address the effect of hydration on shooting performance in adolescent handball players under three conditions: fluid restriction, water intake, and sports drink consumption. Taking into account the physiological and metabolic distinctions between adults and adolescents, the study clarifies the special effects of hydration on male adolescents, a vulnerable but essential group in sports.
The topic is interesting and pertinent to the knowledge of obesity and lifestyle choices. The study simulates actual match conditions while upholding scientific rigor by using a well-controlled setting with constant humidity and temperature. Using validated methods such as the JUGS Gun to measure shooting velocity and accuracy guarantees quantifiable and repeatable results.
Accuracy and shooting speed and are essential performance metrics in handball. Training plans and match tactics can be immediately enhanced by knowing how hydration affects these measures.
Although the study is clearly written, well-organized, and uses an easy-to-follow terminology there are certain places where the methodology might be broadened or enhanced for increased reliability and validity.
I have some comments/suggestions in order to improve quality of the study:
INTRODUCTION
- The Abstract is too long. The authors should respect the requirements specified in the instructions for the authors of Nutrients journal (https://www.mdpi.com/journal/nutrients/instructionsconcerning the maximum length of 200 words for the Abstract (currently it has 220 words);
Author's Response: We thank the reviewer’s for the comment. The abstract was reorganized according to the 200-word criterion determined by the journal in line with the reviewer's suggestion.
- In the Introduction, the purpose of the work and its significance are defined and the three specific hypotheses being tested are included. The authors should clearly specify why they consider that their research is original and relevant to the field and what is added to the subject area compared with other cited materials;
- Author's Response: We thank the reviewer’s contribution, We apologise for the ambiguity. With the reviewer’s suggestion, our aims and hypotheses have been reorganized more understandably.
The shooting accuracy and speed are effects of the handball game results. Therefore, it is important to investigate the parameters (accuracy and speed) affecting shooting performance. For these reasons, this study aimed to investigate the effects of fluid restriction and different types of fluid intake on shooting performance and shooting speed in adolescent handball players.
Main hypotheses of this study; i) fluid restriction during training may impair shot accuracy and speed in adolescent handball players, ii) water and sports drink intake may positively affect shot accuracy and speed, iii) fluid restriction and fluid intake protocols may affect shot accuracy and speed at different levels according to shot performance level.
- The acronyms BM, LR are not defined into the text;
Author's Response: We thank the reviewer’s contribution. In line 27, the BM abbreviations are explained. Line 191 LR abbreviations are excluded.
Body Mass (BM)
- The Note from lines 190-191 seems not finished;
Author's Response: We thank the reviewer’s contribution, In lines 190 and 191 reorganized.
- In line 338, a reference has to be included for” Baker and al.”
Author's Response: We thank the reviewer’s. At line 338, a reference number for ‘Baker et al’ has been added.
Baker et al., sub-elite basketball players investigated the effects of fluid restriction on shooting performance in basketball. In the study, the athletes performed 3-h. intermittent walking exercise on a treadmill followed by free shootings from the foul line. As a result of the study, they found that the participants's shooting accuracy rate deteriorated under the hypohydration condition [8].
- The study should include as limitationsthe following aspects:
- Only male teenage handball players were included in the sample, which limited its applicability to female players. The findings would be more broadly applicable if the authors extended their study to female athletes in the future and looked into any potential variations in the effects of hydration caused by physiological variables, such as hormonal changes, that alter fluid balance;
- The study looks at performance in brief sessions without addressing the long-term effects of hydration practices on skill development or recovery;
Author's Response: Thank you for your valuable contribution on limitations. Limitations related to gender difference, female athletes and prolonged hydration processes have been added to the relevant section.
In this study, fluid deficit was calculated based on the body weight of the participants before the experiment design. One of the limitations of this study is that the fluid intake of the participants could not be monitored and determined during the period before the start of the experiment (i.e. the last week). In future studies, it is important to monitor fluid intake for a longer period before the experimental procedure. Finally, since our research was conducted in a controlled indoor environment and only male adolescents, the findings may not be fully generalised to outdoor conditions (different temperature and humidity ) and female adolescents.
- The description of the references should respect the requirements specified in the instructions for the authors of Nutrients journal (https://www.mdpi.com/journal/nutrients/instructions). The authors should respects this requirement.
- Author's Response: We thank the reviewer’s comment. References are organised in accordance with the rules of the journal.

Round 2
Reviewer 1 Report
Comments and Suggestions for Authors
Accept in present form
Reviewer 2 Report
Comments and Suggestions for Authors
I would like to thank the authors for taking into consideration my comments and suggestions and revising their manuscript in order to improve the clarity, transparency and readability of their paper. The provided explanations clearly described how they have addressed each of my comments and what changes have been made.
The authors took into consideration all my concerns /recommendations.
I consider that the paper can be published.